# Correcting the Estimation of Viral Taxa Distributions in Next-Generation Sequencing Data after Applying Artificial Neural Networks

**DOI:** 10.3390/genes12111755

**Published:** 2021-10-31

**Authors:** Moritz Kohls, Magdalena Kircher, Jessica Krepel, Pamela Liebig, Klaus Jung

**Affiliations:** Institute for Animal Breeding and Genetics, University of Veterinary Medicine Hannover, D-30559 Hannover, Germany; moritz.kohls@tiho-hannover.de (M.K.); magdalena.kircher@tiho-hannover.de (M.K.); jessica.krepel@tiho-hannover.de (J.K.); pamelaliebig@yahoo.de (P.L.)

**Keywords:** artificial neural networks, classification, machine learning, metagenomics, next-generation sequencing, viruses

## Abstract

Estimating the taxonomic composition of viral sequences in a biological samples processed by next-generation sequencing is an important step in comparative metagenomics. Mapping sequencing reads against a database of known viral reference genomes, however, fails to classify reads from novel viruses whose reference sequences are not yet available in public databases. Instead of a mapping approach, and in order to classify sequencing reads at least to a taxonomic level, the performance of artificial neural networks and other machine learning models was studied. Taxonomic and genomic data from the NCBI database were used to sample labelled sequencing reads as training data. The fitted neural network was applied to classify unlabelled reads of simulated and real-world test sets. Additional auxiliary test sets of labelled reads were used to estimate the conditional class probabilities, and to correct the prior estimation of the taxonomic distribution in the actual test set. Among the taxonomic levels, the biological order of viruses provided the most comprehensive data base to generate training data. The prediction accuracy of the artificial neural network to classify test reads to their viral order was considerably higher than that of a random classification. Posterior estimation of taxa frequencies could correct the primary classification results.

## 1. Introduction

Next-generation sequencing (NGS) is now regularly used to identify viral sequences in the biological sample of an infected host in order to relate the presence of a virus with disease symptoms of the host [1,2,3]. Besides detection of individual viruses, NGS data is regularly used to determine the overall taxonomic composition of pathogens for the purpose of comparative metagenomics [4] or to reconstruct networks between microbial taxa [5]. Most computational virus detection pipelines or pipelines for determining the taxonomic composition map sequencing reads or assembled contigs against viral reference sequences available in public or own curated databases [6,7,8,9,10,11,12,13]. These mapping approaches have been proven to be successful in a large number of examples, however, they mostly fail to classify reads from new emerging viruses whose sequences are not yet deposited in a database. Ren et al. [13], for example, mention that novel viruses ‘for which their hallmark genes have not been characterized or are poorly represented in reference databases’ can be missed. The difficulty of identifying novel viruses when also no homologues are detectable in sequence databases was pointed out by Maclot et al. [14]. Parks et al. [15] argue that an effective taxonomic assignment algorithm ‘must deal explicitly with unrepresented lineages, either by mapping such novelty to an appropriate taxonomic level [...] or by using a criterion that allows some reads or assemblies to remain unassigned’. While there are diverse tools available for taxonomic classification of NGS reads from novel species, these share several limitations. For instance, the Kraken-1 tool [16] matches *k*-mers (with *k* up to 31) with a reference of genome sequences; however, the sensitivity drops to 33% in the best case if the species is yet unknown. As the authors state, this is due to ‘Kraken’s reliance on exact matches of relatively long *k*-mers: sequences deriving from different genera rarely share long exact matches’. Despite improvements of the metagenomic data analysis algorithm, Kraken-2 [17] still shows low sensitivity on novel viruses.

Besides classification approaches by mapping, machine learning approaches have been used to classify reads to taxonomic ranks [18] such as genus, family, order, class and phylum. Among the machine learning approaches listed by Peabody et al. [18], only one approach studies the performance of deep learning models for taxonomic classification (Rasheed et al. [19]). Rasheed et al. use artificial neural networks (ANN) to classify reads to different taxonomic ranks and evaluate their models on datasets representing about 500 microbial species. Some approaches have also been undertaken to use machine learning models on NGS data to identify or classify reads that belong to novel emergent viruses. Zhang et al. [20] as well as Bartoszewicz et al. [21] employ diverse machine learning models to identify viruses that can infect humans. Whilst Zhang et al. applied the k-nearest neighbour (k-NN) model using *k*-mer frequencies as input features for their classification, Bartoszewicz et al. employed a convolutional neural network (CNN) on simulated reads and contigs.

In this work, in contrast to the approaches by Zhang et al. and Bartoszewicz et al., the ability of an ANN to classify sequencing reads on the taxonomic level of nine biological orders was studied. Although the main focus of this work is on ANN models, we include linear discriminant analysis (LDA) and support vector machines (SVM) for comparison. The level of biological order was chosen since the data base is best suited among the different taxonomic levels, while the ANN studied by Rasheed et al. was fitted on sequence reads from a rather small number of microbes. We trained our model on the basis of several thousand viral sequences available at the NCBI database [22]. In comparison to SVMs or LDA, ANNs are reported to achieve the highest accuracy rates on test data, although it depends on the specific classification problem [23,24]. However, despite being trained on a weak data base, an ANN can still produce a large number of misclassifications. Therefore, we additionally incorporated auxiliary test sets of sequencing reads with known taxonomic memberships and usde these to estimate conditional class probabilities in the individual taxonomic orders. These estimates were then used to correct estimated taxonomic distributions in the actual test sample. For that correction, we present new formulas to derive posterior class distributions.

Before fitting ANNs, we linked available viral reference sequences with their taxonomic description and studied the distribution of different taxonomic levels (phyla, subphyla, classes, subclasses, orders, etc.) to identify which of these ranks a solid data basis for building a training dataset would be reasonable. Next, we extracted *k*-mer distributions and inter-nucleotide distances as input features from the available reference sequences. In total, we generated 120 input features, and studied which combination qualified best for fitting an ANN.

Feature selection, building of the training data, the structure of the ANN model as well as the correction methods are provided in the following methods section. In the subsequent results section, a simulation study in which we cross-validate the performance of the ANN and the estimation of the taxonomic frequency distributions is presented. In addition, we compare the precision of the estimations with frequency distributions derived from a mapping approach. Finally, we discuss benefits and limitations of our approach.

## 2. Materials and Methods

### 2.1. Linking of Taxonomy and Genome Data

The taxonomy database containing data needed for taxonomic classification was downloaded from the ftp-server (ftp://ftp.ncbi.nih.gov/refseq/release/viral, accessed on 15 March 2020) of the National Center for Biotechnology Information [25] in March 2020. An extract of the taxonomy database file fullnamelineage.dmp from this ftp-server is depicted in Figure 1. In this file, taxonomic information about various species are listed in rows. Beginning with archaea, bacteria and fungi, viral data is included from line 2,007,414 onwards. Row entries contain species names and taxonomy levels which are, however, neither labelled nor complete. We assigned these taxa to taxonomic levels based on the classification scheme of the International Committee on Taxonomy of Viruses [26]. To enable correct assignments, we used the common taxa endings as specified by the International Committee on Taxonomy of Viruses [26]. e.g., the ending ‘virus’ indicates the realm, the ending ‘vira’ the subrealm, the ending ‘virae’ the kingdom, etc. In Figure 1, for example, the Bovine adenovirus 4 on line 2,007,418 is a member of the family of Adenoviruses.

As the taxonomic database does not provide any unique NC identifier, the species names of the reference genome file were searched in the taxonomy file to link genomic with taxonomic data. We obtained viral reference genomes as FASTA-files [22] again from the NCBI database and concatenate the three files into a single one. By combining genomic and taxonomic data, we created a structured taxonomy table including FASTA-files of species which replaces the unstructured original taxonomy file. As the original files contain missing data, we only chose species with available taxonomy information. From more than 2 million taxonomy entries, only about 200,000 entries remain that are related to viruses. In contrast, the FASTA-file contains only 12,180 viral genome sequences and 2332 sequences provide taxonomy data. Hence, a maximum of 2332 viruses can be used in the training data.

### 2.2. Selection of a Suitable Taxonomic Category

From a list of 14 taxonomy levels (Table 1), one needs to decide which one constitutes a suitable basis as a training set. As a necessary criterion, any classification algorithm requires at least two different taxonomy groups. Furthermore, to achieve a high accuracy, artificial neural networks need a large amount of training data in relation to the number of different groups one wants to predict. As a consequence, we looked for a taxonomic category that has more than one, but not too many different levels and that provides a sufficient amount of available viruses per level. e.g., the two taxonomic levels ‘family’ and ‘genus’ have a large number of annotated viruses, but too many different groups, each. The level ‘suborder’ would have a comfortable number of 5 groups, but too few annotated viruses. This is why we chose the taxonomy level of ‘orders’. Viral orders and the numbers of stored species per order in the NCBI database are *Bunyavirales* (7), *Caudovirales* (1859), *Herpesvirales* (22), *Ligamenvirales* (21), *Mononegavirales* (70), *Nidovirales* (47), *Ortervirales* (81), *Picornavirales* (107) and *Tymovirales* (118). Take note, that the distribution is very unbalanced between the different order groups. For example, there are only seven viruses in the order *Bunyavirales*, but 1859 viruses in the order *Caudovirales*.

### 2.3. Sampling of Viral Reads for Training and Test Sets

Next, we detail how viruses and reads were sampled from which the input features are computed subsequently. We first used the FASTA-file provided by the NCBI database, restricted to 2332 viral genome sequences whose taxonomy data are available. From each order $j\in \{1,...,N_{cat}=9\}$, we randomly sampled the corresponding viruses without replacement and divided them randomly into training and test viruses. For optimizing the weights in an ANN, the training data was typically split into training and validation set. The disjunct partition of training, validation and test data was chosen as 70%, 15% and 15%, respectively. For example, from seven *Bunyavirales* viruses, five were used for training, one for validation and one for the test data (Table 2).

In step two, we randomly sampled with replacement the same amount of viruses out of each of the 3×9=27 cells in Table 2. I.e., from each order $j\in \{1,...,N_{cat}\}$ and for training, validation and test data separately (i∈{1,2,3}), we sampled $n_{i,j}$ of $N_{i,j}$ viruses which yielded a total amount of *n* viruses: $\forall j\in \left\{1,...,N_{cat}\right\}:n_{1,j}+n_{2,j}+n_{3,j}=0.7n+0.15n+0.15n=n_{.,j}=n.$

The whole sampling plan from step one and two is also known as two-stage stratified sampling with disproportionate allocation [27]. After sampling the viruses, we sampled reads of the FASTA file by choosing random start positions out of a discrete uniform distribution over each viral genome and a read length of 150, a typical reads lengths in NGS.

### 2.4. Specification and Selection of ANN Input Features from Sequencing Reads

Possible input features to select for an ANN are for example *k*-mers [28], inter-nucleotide distances [29] and DNA sequence motifs [30]. A *k*-mer is a subsequence of length *k* of the original read sequence. Inter-nucleotide distances are the distances between two identical bases. For instance, the DNA sequence TTATACTACGTGGGGGGGGGTCCT exhibits T-T distances of 1, 2, 3, 4, 10 and 3 and we are interested in the count frequencies of the distances 2 to 10. A DNA sequence motif is a set of words which can have arbitrary or a subset of A, C, G and T letters on particular base positions while having fixed letters on other positions. Here, we focus on DNA sequence motifs with two or three fixed letters, therefrom two at the margins, such as C.....T or G..G...A.

The overall number of input features studied in our analysis, sums up to $4^{1}+4^{2}+4^{3}+4\times 9+x+y=4+16+64+36+x+y=120+x+y,$ where $4^{k}$ are the numbers of *k*-mers, 36 inter-nucleotide distances and x=19 and y=24 the numbers of selected DNA sequence motifs with two or three fixed letters, respectively. As we will describe below, no DNA sequence motifs were selected. Therefore, 120 input features were finally studied. We did not include 4-mers or DNA sequence motifs with four fixed letters because this would result in too many input features and the probability of obtaining frequencies equal zero was too high.

Selection of input features was based on the results of Kruskal–Wallis tests and by accuracy comparisons in additional ANN simulations. Separately for each sequence motif, the relative frequency in each complete nucleobase sequence of the FASTA-file was counted. Next, the frequencies between and within the groups or orders were compared by Kruskal–Wallis tests [31]. High differences of the frequencies between the orders in relation to low differences within the orders point out that this particular DNA sequence motif would be a good input feature to differentiate between the orders. Finally, we selected those sequence motifs with the lowest *p*-value and highest effect size $\epsilon^{2}$ [32], respectively, up to a sequence length of eight. From all possible DNA sequence motifs with two or three fixed letters, we chose those with an effect size not less than 0.08 or 0.2, respectively.

We iterated 10 simulation runs per layer in the artificial neural network and evaluated mean and standard deviation of accuracy results (see Table 3). Each layer uses a subset of all proposed input features. If the paired *t*-test recognises a statistically significant and relevant accuracy increase of the subsequent layer, its corresponding input features are added to the ANN model. As a results, the feature selection method advocates to use 1-mers, 2-mers, 3-mers and inter-nucleotide distances as input features.

### 2.5. Artificial Neural Net Structure

For computation of the ANN [33], we applied the R-package keras [34]. The sequential model we used contains 100,000 training reads per virus order. The input layer contains 120 input nodes, the hidden layer 64 nodes and the output layer nine nodes representing the nine different orders. The activation functions used were ‘Rectified Linear Unit’ (ReLU) [35] for the hidden layer and ‘Softmax’ [36] for multinomial classification of the output layer. As an optimisation algorithm we chose ‘stochastic gradient descent’. To quantify the model performance, sparse categorical crossentropy and accuracy were used. In order to study whether a more complex ANN can reach more accuracy, we also included a model with 5 hidden layers, each again with 64 nodes and the ReLU activation function.

While training the model, the data were shuffled before each epoch to allow for faster convergence and to prevent memorizing of the order of the training samples [37]. Batch sizes of $2^{10}$ were used as well as a maximum number of training epochs of 500 as an early stopping method to prevent overfitting [38].

### 2.6. Building of the Confusion Matrix and Measures of Classification Performance

The typical summary table of classification results is a confusion matrix. In our case, cell entry (j,k) of the confusion matrix provided the number of reads of true order *j* classified to order *k* ($j,k\in \{1,...,N_{cat}=9\}$). In our particular test data sets, 21,429 reads per order were given, represented by the row-sums of the confusion matrix. The diagonal from top left to bottom right provides the correct classification frequencies from which the accuracy can be derived.

As additional performance metrics, the true positive rate (TPR, sensitivity) and true negative rate (TNR, specificity), as well as positive and negative predictive values (PPV and NPV) can be reported. These four measures must be interpreted in the sense of one biological order versus the eight other orders, here.

### 2.7. Estimation Correction of Taxa Frequencies

The artificial neural network classifies single reads into taxonomic orders. However, we do not only want to predict to which order a particular read belongs, but also want to estimate the global taxa frequency distribution of the nine orders in a biological sample. In this way we gain a summary of species abundance on taxonomic levels. The number of reads assigned by the ANN to the nine orders provides a prior frequency distribution. Since the sequences of viruses within one order can have little similarity, a good classification performance of an ANN is still a challenge, and a large number of misclassifications and thus a biased prior estimation of taxa distributions is likely. Therefore, we employed additional auxiliary test sets of labelled reads to determine conditional probabilities that a read classified to a particular order belongs to this one or to one of the other orders. We then used this additional information to estimate a posterior, less unbiased distribution. The correction procedure is described in the following (cf. Figure 2).

We denoted the number of auxiliary data sets as $N_{sim}$ and the number of taxonomic categories as $N_{cat}$; in our scenario the categories are the 9 viral orders. Each of the $N_{sim}$ auxiliary data sets consists of a training and a test sample. The training sample was further split into training and validation set to fit the parameters of the ANN. The fitted ANN was then applied to the auxiliary test sets resulting in $N_{sim}$ confusion matrices:$$C_{s}\in N_{0}^{N_{cat}\times N_{cat}},s=1,\cdots ,N_{sim},$$
with true order labels in rows and predicted labels in columns. The column sums from the matrices $C_{s}$ compose a prior estimation
$$\hat{A}\in N_{0}^{N_{cat}\times N_{sim}}$$
of the read count distribution in the nine orders. Furthermore, the confusion matrices $C_{s}$ were used to calculate the matrices
$$P_{s}\in {\left[0,1\right]}^{N_{cat}\times N_{cat}},s=1,\cdots ,N_{sim},$$
which contain the conditional class probabilities that given a read classified to order $k^{*}$, thus it truly belongs to order $j^{*}$. From the matrices $P_{s}$ leave-one-out mean matrices
$${\bar{P}}_{s},s=1,\cdots ,N_{sim}.$$
can be calculated using all matrices $P_{s}$ except the one corresponding to the particular simulation run.

By applying the law of the total probability and multiplying each leave-one-out mean matrix ${\bar{P}}_{s}$ by the related *s*-th column of $\hat{A}$, we receive the posterior estimations of read count distributions for the auxiliary test samples *s*:(1)$$\hat{B}\in R_{\geq 0}^{N_{cat}\times N_{sim}};{\hat{B}}_{.,s}={\bar{P}}_{s}\times {\hat{A}}_{.,s}.$$
i.e., in test run *s* the posterior probability that a read in the sample belongs to order *j* is given by the sum of conditional probabilities of order *j* multiplied by the prior probabilities of orders *k* in run *s*:(2)$$Prob_{s}\left(j\right)=\sum_{k=1}^{N_{cat}}{\bar{p}}_{s,jk}\times {\hat{a}}_{k,s}.$$

Thus, the read count distribution ${\hat{A}}_{.,s}$ of run *s* is corrected by information about the conditional class probabilities estimated from the independent s−1 other simulation runs.

When analysing a real-world test sample with unknown class memberships, matrices ${\bar{P}}_{s}$ would be obtained from the above simulations with auxiliary test sets and the classification of sequencing reads from the real world sample would result in a new matrix $\hat{A}$ with dimension $N_{cat}\times 1$. Since $N_{sim}$ test samples were generated artificially, the corrected (i.e., posterior) distribution of taxa $\hat{B}$ would instead have dimension $N_{cat}\times N_{sim}$.

## 3. Results

Our primary research question was to study the ability of an ANN to classify sequencing reads with unknown taxonomic order and to subsequently deriving the frequency distribution of these orders in the sample. First, we present the achieved accuracy of our fitted models, and demonstrate then how our correction approach can be used to obtain an improved estimation of the taxonomic order frequency distribution. The validation of our approach was performed on simulated sequencing data with $N_{sim}=10$ runs. In each run, different sets of viruses and viral reads were drawn. Finally, we demonstrate the effect of the correction approach in a publicly available sequencing sample taken from a harbour seal.

### 3.1. Performance History of the Network Fitting Processes and Results from Other Machine Learning Models

Exemplarily, the performance history of 4 out of 10 simulation runs is depicted in Figure 3 in form of the loss and accuracy with respect to the training epoch of the ANNs with on hidden layer. Accuracy and loss curves are presented for training and validation data. We allowed the maximal number of epochs to be 500, however, in some runs, the fitting converged much faster. Training loss decreased negatively exponential and training accuracy increased logarithmically. After the final number of epochs was reached, validation accuracy was considerably higher than 11% in all ten runs. Considering that we have nine taxonomic orders, this means that the ANN classifications were at least more precise than random classifications. Plot A shows an exceptional validation performance behaviour. Validation loss decreased slightly until epoch 150, then increased to a value higher than at the start, while validation accuracy was still increasing. This indicates that the uncertainty for predicting class probabilities increases, whereas the classification results are becoming more precise. Plots B to D partly show better loss and accuracy performances on validation data in comparison to training data.

The confusion matrix in Table 4 shows true (rows) versus classified (columns) orders. Presented counts are averaged over the 10 simulation runs. For each order *j*, classification results were additionally treated as results from a binary classifier, i.e., order *j* versus all other orders. Accordingly, sensitivity (i.e., true positive rate, TPR), specificity (true negative rate, TNR), positive (PPV) and negative predictive values (NPV) were derived. Averages of these four measures over all 10 simulation runs are shown in Table 5, while TNRs and NPVs are high for all orders, TPRs and PPV are rather weak.

Although achieved accuracies and the other measures of performance were better than by a random classifier, the fitted ANNs still resulted in a large number of misclassifications. Consequently, the estimated frequency distributions for the nine biological orders are biased. These results were for us the reason to develop the correction procedure detailed in the methods section. Simulated findings for this correction procedure are provided in the next subsection.

The mean accuracy of the simple ANN model to correctly classify a read as derived from the 10 simulation runs was 0.40 (minimum: 0.35, maximum: 0.46.), which is higher than a random classifier and would result in an accuracy of 0.11, due to the nine order. We also checked whether a more complex ANN with five hidden layers, or other machine learning approaches, explicitly LDA or SVMs with different kernels, can improve the accuracy, while the ANN with five hidden layers and the SVM with polynomial kernel achieved nearly the same (but also no better) accuracy than the ANN with only one layer, the other models achieved clearly worse accuracies (Table 6).

### 3.2. Prior and Corrected Estimation of Taxa Frequencies

Boxplots in Figure 4A show the distributions of mapping, prior and posterior (i.e., corrected) estimation of taxa frequencies of the nine biological orders obtained in 10 simulation runs for a true balanced taxa distribution, i.e., with 1/9∼11% of reads per order in training, validation and test set, and when using an ANN with one hidden layer. Both, variance and bias, and thus mean squared error of posterior estimation, are considerably smaller for each order in comparison to prior or mapping estimation. As was seen with the accuracy in the previous section, no clear improvement of the posterior estimation can be observed when fitting a more complex ANN model with five hidden layers (Figure 4B).

Furthermore, frequency estimation by the mapping approach performed worst among the competing approaches. In general, the mapping rates were low with a mean value of 14.5% and a standard deviation of 3.3% which can be traced back to the fact the viral reads in the test data were disjunct to those in the training data. Among the mapped reads, not a single one was mapped to a virus from the orders *Bunyavirales* or *Tymovirales*, and consequently taxa frequencies of these orders were underestimated. In contrast, *Caudovirales* and *Ligamenvirales* orders were overestimated considerably. On the median level, more than 40% of all reads were mapped to *Caudovirales* although only 100/9≈11% were expected. Since the difference in the statistical estimation error of the mapping approach in comparison to the ANN approaches is graphically obvious, no statistical test was performed to proof its inferiority.

Besides the graphical comparison, we applied Mood’s median test to compare the median absolute deviation of estimated frequencies from the true value of 21,429 (i.e., 11%) test reads between the prior and posterior estimation. The resulting *p*-value is <0.0001 which is significant when taking an α of 5%. Median absolute deviations of prior and posterior estimation were 8872.5 or 2741.1 reads, respectively, which translates to a difference of 6131.4 reads. Thus, median posterior estimation of taxa frequencies is over 6000 reads more precise than median prior estimation.

In order to see whether the correction approach also works for other machine learning approaches than ANNs, we applied it to the SVM with polynomial kernel and LDA, while there are clearly different prior estimations obtained from SVM and LDA, the posterior estimation again is able to correct this bias from the prior estimations (Figure 5). Corresponding to the low accuracy for read classification, the bias of the prior taxa estimation with LDA is also much worse than that of the SVM and the ANN.

While the above scenarios where based on the same distributions of viral orders in the training, validation and test sets, we also simulated more realistic scenarios with different taxa distributions in these data sets. In particular, the distributions for training and validation set were kept balanced (i.e., 11% of reads per set), while the distributions of the test sets where drawn randomly from the Dirichlet-multinomial distribution [39]. In a real-world scenario, the analyst would not be restricted to use other than balanced distribution for training and validation set; however, a true test set from a biological sample would most likely show an unbalanced distribution of the nine orders. We simulated two of such example scenarios and fitted again ANNs with one hidden layer. In both scenarios, there is a clear bias by the prior estimation, while the posterior estimation corrects this bias well (Figure 6).

### 3.3. Effect of Frequency Correction in Sequencing Data from a Harbour Seal

To demonstrate how biased and corrected frequency estimation can affect the biological conclusion in a real sample, we applied the machine learning approach on a FASTQ-file generated from a biological sample of a harbour seal [40]. In this real-world sample, we do not know the true distribution of orders, thus this example can only demonstrate the shift between prior and posterior estimation. After mapping to a database of viral reference genomes [22] and filtering of low quality reads with a mapping quality smaller than 2, 3,154,562 sequencing reads remained for classification and estimation of taxa frequencies. Following the procedure depicted in Figure 2, we first received prior estimations of order frequencies, and used $N_{sim}=10$ auxiliary data sets to determine conditional classification probabilities $P_{s}$. The latter information was used to correct prior estimations according to formula (1). In contrast to the above simulation, only one frequency value per biological order is obtained for the prior estimation (Figure 7). Due to the 10 auxiliary data sets posterior results provide a frequency range for each order, while the prior estimation yields nearly zero percent frequency for 5 of the studied orders, correction shows a slightly changed distribution, which could have a strong impact on the biological interpretation of a sample within a metagenomics analysis.

## 4. Discussion

### 4.1. Performance Comparison between Machine Learning and Mapping Approach, and Correction Approach

In this work, we studied the ability of an ANN models, SMVs and LDA to classify viral sequencing reads to nine different biological orders. Other approaches that have so far studied machine learning models for classification of sequencing reads concentrated on microbial species [19], or in the context of viruses on models to distinguish human pathogenic from non-pathogenic viruses [20,21]. After screening the availability of training sequences by different taxonomic levels, we identified the level of order as the most comprehensive basis to fit a machine learning model. We found that the ANN models and an SVM with polynomial kernel performed clearly better than a mapping approach which can not cope with unknown viruses, and also better than LDA and SVMs with other kernels. However, while we could observe an overall accuracy that were considerably higher than a random classification, there was still a large number of misclassifications and consequently a biased frequency distribution of the nine orders. Several factors can be considered that might improve the performance of the ANN model, such as length of the sequencing reads, the selected input features derived from the training reads, and the sample size of the training data. Moreover, the parameters of the ANN (e.g., number and size of hidden layers and activation functions) could result in an improved performance. However, even with optimized training data and ANN configuration, the large heterogeneity of viral sequences within orders as well as a rather small heterogeneity between orders makes the classification difficult.

Although the overall performance of the best models in our evaluation was weak, the estimation of order frequencies could be helpful to support or complement the findings of a mapping approach by classifying reads from novel viruses. One possibility could for example be to apply first a mapping approach, and all non-mapped reads are subjected to a machine learning model.

For fitting the machine learning models, we draw training reads from only 70% of viruses from each order. With this, we took into account scenarios with reads of novel viruses in the data of a test sample. If an machine learning model was trained on reads from all available viruses in a database instead, we would expect that a mapping based approach would excel the machine learning model performance. Nevertheless, the machine learning model model could then be used to validate the mapping results. Independently of which of the two approaches, mapping or machine learning, are used for taxa classification, in both cases taxa abundance estimations could be enhanced by the posterior estimation formulas we presented. For the approach of correcting the abundance estimations, we make use of the possibility that one can draw any number of independent data sets of labelled reads from the NCBI database, and to use these ‘auxiliary’ data sets to assess the conditional class probabilities. Instead of having one large training set, the additional data sets can account for the variability of the conditional class probabilities.

An improvement that can be considered for our approach, and as was requested by Parks et al. [15], would be to allow the ANN or other machine learning models to assign a read to none of the nine viral orders. One possibility would be to generate random reads for the training data that don’t belong to any known virus of the nine orders. It is not straightforward how the sequences of such reads should be composed, but we will follow this question in our future research.

### 4.2. Estimation of Real-World Data Taxa Abundances

Prior and posterior estimations of taxa abundances of the harbour seal sample in Figure 7 show that prior estimations were corrected in a different amount. For instance, the relative frequency of *Tymovirales* classifications was scaled down to a much higher degree than the abundance of *Ortervirales* sequences. A possible reason for the high number of read sequences classified as *Tymovirales* is that according to Table 4 that a relatively large number of reads of other orders are falsely classified as *Tymovirales*, but not the other way around. This increases the risk that reads are falsely classified into the *Tymovirales* order which is why this frequency is scaled down on posterior estimation.

On the other hand, *Picornavirales* is scaled up massively, in contrast to other orders which have a very low prior estimation, too, but are scaled up to a much lower degree. This is due to the fact that *Picornavirales* exhibit the lowest true positive rate (see Table 5) and thus for the posterior estimation it can be assumed that a vast amount of *Picornavirales* reads could not be detected initially by the ANN classification why this frequency is scaled up considerably.

Although the true taxonomic orders of the read sequences of the harbour seal sample are not known, the orders of viruses reported by Rosales and Thurber [40] were found by our estimation procedures, too. Viral genera associated with stranded harbour seal brain tissues were *Cripavirus*, *Mardivirus*, *Simplexvirus* as well as *Varicellovirus*. With a few exceptions most of these sequence counts origin from *Varicellovirus* genus which is allocated to *Herpesvirales* order. The same applies to *Mardivirus* and *Simplexvirus*. *Cripavirus* which is only detected in one sample in a low amount belongs to order *Picornavirales*.

### 4.3. Phylogenetic Explanations for Misclassifications

In our evaluation of the neural net for classification of viral orders, several misclassifications were observed, while according to Table 5 the relative majority of reads from most orders is correctly classified, but too many reads are classified to the wrong order. Due to the partial similarities between *Bunyavirales* order and *Paramyxoviridae* family [41], which belong to *Mononegavirales* order, most of *Bunyavirales* reads are falsely classified into *Mononegavirales* order, but not the other way around. On average, 7606 *Bunyavirales* reads are classified to *Mononegavirales* and only 2728 *Mononegavirales* reads are classified to *Bunyavirales*. This is justified by basic principles of set theory. Since *Bunyavirales* and *Mononegavirales*, to which the *Paramyxoviridae* family belongs, share the same ancestor, a relative majority of *Bunyavirales* reads is classified to the subset of *Mononegavirales* viruses that belong to the *Paramyxoviridae* family and thus is classified correctly. Contrariwise, only *Mononegavirales* reads selected from *Paramyxoviridae* family are majoritarian classified to *Bunyavirales* order and this does not apply for reads from other *Mononegavirales* families. Another reason for the false classification of reads from *Bunyavirales* to the order *Mononegavirales* could be explained by the close phylogenetic relation between these two orders (cf. Figure 1 from Wolf et al. [42]).

*Nidovirales*, *Picornavirales* and *Tymovirales* are biologically related and *Herpesvirales* descend from *Caudovirales* which explains high misclassification rates between these orders (see Table 4). For instance, 3816 or 4458 *Picornavirales* reads are mapped to *Nidovirales* or *Tymovirales*, respectively. Furthermore, the phylogenetic relationship between *Picornavirales* as an ancestor and *Nidovirales* as a descendant on branch two [42], which both belong to positive-sense RNA viruses, is somewhat close which results in a high one-way misclassification rate from *Picornavirales* to *Nidovirales*. *Herpesvirales* originate form *Caudovirales* [43,44] which explains the high two-way misclassification rates between these orders.

## 5. Conclusions

Besides mapping approaches or homology comparisons, machine learning approaches are regularly used to classify sequencing reads to taxonomic units in meta-genomics analyses, while taxonomic classification is more common for the analysis of microbiomic communities, it is less extensively used for the analysis of viromes. Here, we identified that the taxonomic level of viral order provides a solid basis to build training data for machine learning models, because sufficiently viruses are annotated for this level in public databases. We observed that accuracies for read classification are better than a random classification, however, still many miss-classifications are observed. Consequently, prior estimations of taxa distributions are biased. We proposed an approach that uses additional test sets generated from the sequence database to assess conditional probabilities that a read is classified to class *j* given that it belongs to class *k*. We have shown in several scenarios that the prior taxa estimations can be corrected based on these conditional probabilities. Therefore, even with low accuracies on the read level, our approach allows for an improved estimation of taxa compositions in virome samples.

## Figures and Tables

**Figure 1 genes-12-01755-f001:**
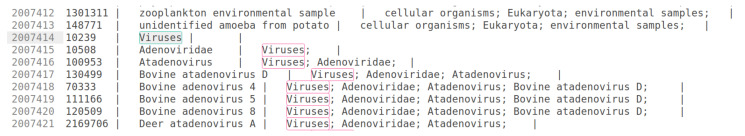
Extract of the virus taxonomy database showing how taxonomic levels are deposited. For many viruses, not each of the taxonomic levels is available.

**Figure 2 genes-12-01755-f002:**
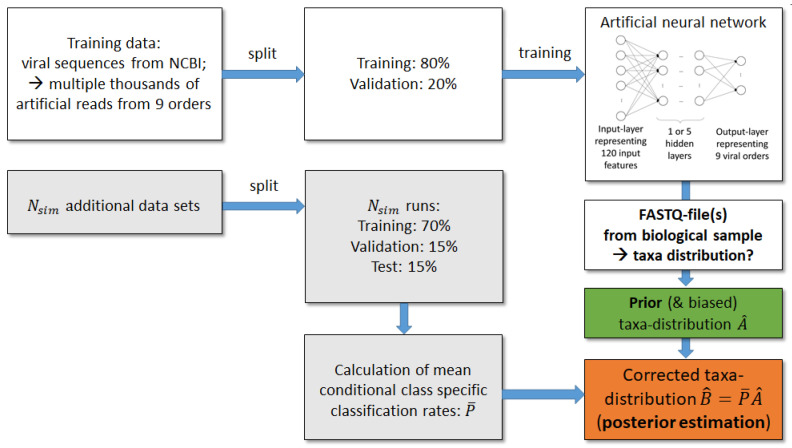
Workflow of estimating taxa distribution from a biological sample. Each ANN was trained with artificial viral sequencing reads drawn from viral reference genomes of the NCBI database. In order to train the parameters of each ANN, these sequences were split into 80% training and 20% validation data. The trained ANN model was applied to the reads of the FASTQ file from the biological sample to obtain a prior taxa distribution $\hat{A}$. To correct this biased taxa distribution, simulation runs on $N_{sim}$ additional auxiliary data sets were performed (grey boxes). Misclassification rates from the additional simulations are taken to correct the prior estimation of frequency distribution.

**Figure 3 genes-12-01755-f003:**
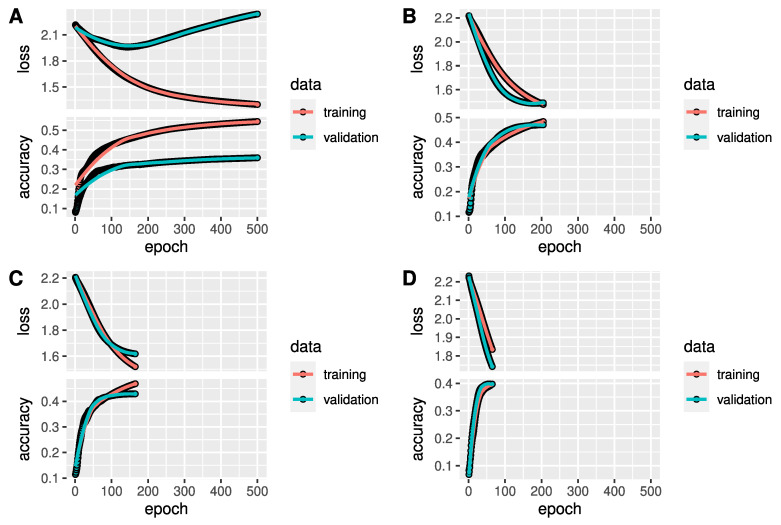
Accuracy performance history. The performance history of four exemplarily chosen simulation runs is depicted. The smoothness of the curves could be reached by a large batch size. The training time varies vastly, exceeding 500 epochs on plot (**A**) and falling below 50 on plot (**D**). Whilst training loss decreases subject to exponential decay, training accuracy increase is logarithmic. Validation accuracy exceeds simple random classification probability of 11% considerably. Plot (**A**) holds a remarkable behaviour of validation performance. From epoch 150 on, validation loss and accuracy both increase simultaneously. This portends that predictions of class probabilities incline to become increasingly imprecise, whereas classification results are increasing in accuracy. In contrast to plot (**A**), on plots (**B**–**D**) loss and accuracy performance on validation data are superior to that in training data.

**Figure 4 genes-12-01755-f004:**
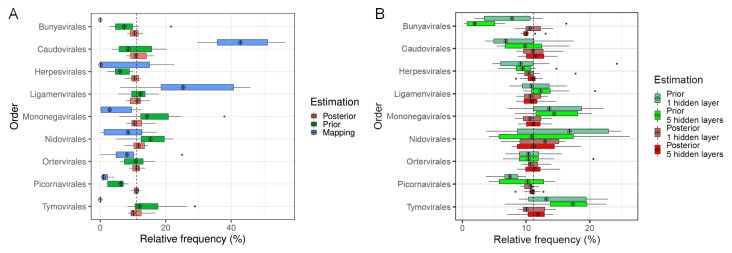
Comparison of estimated taxa frequencies. Distribution of taxa frequencies summarized from 10 simulations runs. (**A**): comparison of estimation by the mapping approach, ANN (1 hidden layer) based prior and posterior estimation of taxa frequencies show that posterior estimations are much closer to the true frequencies of 11% (dashed line) per order in terms of the mean absolute deviation (MAD) and thus more precise in relation to the prior and mapping based estimations. (**B**): comparison of prior and posterior estimations when using an ANN with either 1 or 5 hidden layers.

**Figure 5 genes-12-01755-f005:**
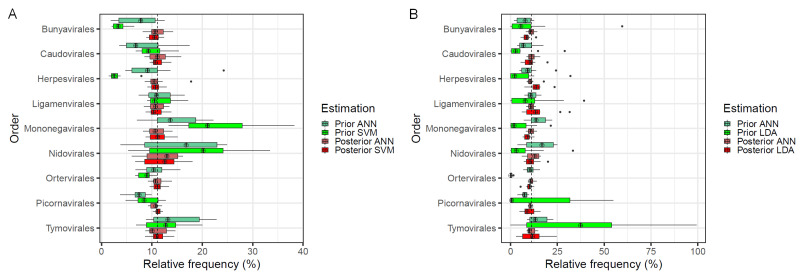
Comparison of estimated taxa frequencies. Distribution of taxa frequencies summarized from 10 simulations runs.
Comparison of estimation by an ANN based prior and posterior estimation of taxa frequencies versus SVM (**A**) or versus
LDA (**B**) based estimation.

**Figure 6 genes-12-01755-f006:**
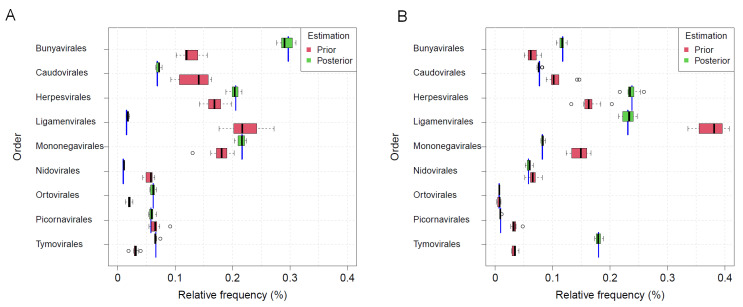
Comparison of estimated taxa frequencies, when there were unbalanced distribution of the reads in the nine viral
orders. In the two example cases (**A**,**B**), the distributions of orders were drawn randomly from a Dirichlet-multinomial
distribution. The blue lines mark the true frequency of each order.

**Figure 7 genes-12-01755-f007:**
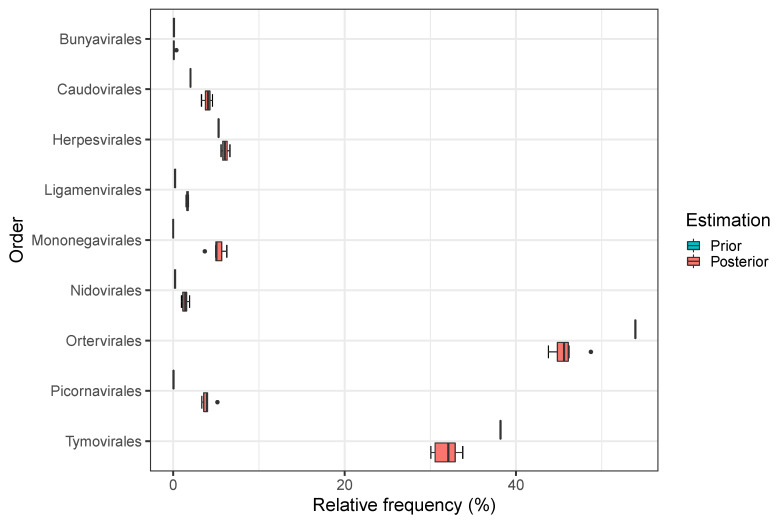
Taxa abundances of a harbour seal. While prior estimations of *Bunyavirales*, *Ligamenvirales*, *Mononegavirales*, *Nidovirales* and *Picornavirales* were very low, posterior estimations scaled up these taxa abundances. For *Mononegavirales* and *Picornavirales* this correction gap is large, whereas for *Bunyavirales*, *Ligamenvirales* and *Nidovirales* it is only small. In contrast, the frequency estimations of *Ortervirales* and *Tymovirales*, which were classified numerously by the ANN, were scaled down.

**Table 1 genes-12-01755-t001:** In total, 14 taxonomic levels are available. For fitting a machine learning model, a moderate number of groups and a fair amount of available viruses per group builds a reasonable training set. Therefore, the taxonomic level of orders was chosen for the evaluation in this work.

Taxonomy Level	Groups	Available Viruses	Available Viruses/Group	Not Available Viruses
Domain (=Viruses )	1	3747	3747	0
Realm	1	883	883	2864
Subrealm	0	0		3747
Kingdom	0	0		3747
Subkingdom	0	0		3747
Phylum	1	77	77	3670
Subphylum	2	77	38.5	3670
Class	2	77	38.5	3670
Subclass	0	0		3747
Order	9	2332	259.1	1415
Suborder	5	47	9.4	3700
Family	100	3482	34.8	265
Subfamily	57	921	16.2	2826
Genus	559	2511	4.49	1236

**Table 2 genes-12-01755-t002:** Numbers of viruses in training, validation and test set. Step one of the sampling procedure was to randomly sample viruses and viral reads from each order without replacement and allocate them to training, validation and test set. A total of 70, 15 or 15% of sampled viruses are used for training, validation or test data, respectively.

	Training	Validation	Test	Σ
*Bunyavirales*	5	1	1	7
*Caudovirales*	1301	279	279	1859
*Herpesvirales*	16	3	3	22
*Ligamenvirales*	15	3	3	21
*Mononegavirales*	50	10	10	70
*Nidovirales*	33	7	7	47
*Ortervirales*	57	12	12	81
*Picornavirales*	75	16	16	107
*Tymovirales*	82	18	18	118
Σ	1634	349	349	2332

**Table 3 genes-12-01755-t003:** Overview on input features for the artificial neural networks. Different combinations *k*-mers and sequence motifs were evaluated according whether they have significantly different distributions in the nine viral orders. Mean accuracy over 10 simulation runs and standard deviation for the different feature combinations are given.

Feature Selection	Number of Input Features	Mean Accuracy	Standard Deviation
1-mers	4	0.303	0.032
1-mers, 2-mers	20	0.380	0.041
1-mers, 2-mers, inter-nucl. dist.	56	0.425	0.049
1-mers, 2-mers, inter-nucl. dist., seq. motifs 2	75	0.424	0.040
1-mers, 2-mers, inter-nucl. dist., seq. motifs 3	80	0.436	0.050
1-mers, 2-mers, inter-nucl. dist., 3-mers	120	0.501	0.025

**Table 4 genes-12-01755-t004:** Average confusion matrix. Confusion matrix showing classification results for 21,429 simulated viral reads, where rows represent the true biological order and columns represent order predicted by the artificial neural network. Results are averaged over 10 simulation runs. Reads on the diagonal were correctly classified, while correct classification rates are higher than those of a random classifier, still a large number of misclassifications occur.

Order	Order ID	Ord. 0	Ord. 1	Ord. 2	Ord. 3	Ord. 4	Ord. 5	Ord. 6	Ord. 7	Ord. 8
*Bunyavirales*	0	5321	190	184	2665	7606	2471	1293	596	1104
*Caudovirales*	1	778	8782	2673	2108	1270	1960	1350	512	1996
*Herpesvirales*	2	480	4950	6193	1808	1706	1682	1407	752	2453
*Ligamenvirales*	3	1271	1635	488	12,071	1451	2099	1233	491	691
*Mononegavirales*	4	2728	497	258	839	9669	1615	2668	949	2206
*Nidovirales*	5	1068	855	414	740	1636	12,134	681	1164	2737
*Ortervirales*	6	1510	999	1015	1391	3859	1013	8198	673	2771
*Picornavirales*	7	1622	1157	710	1046	3067	3816	2047	3504	4458
*Tymovirales*	8	1295	1231	405	315	2770	1730	1731	1176	10,775

**Table 5 genes-12-01755-t005:** Measures of classification performance. True positive and true negative rates (TPR and TNR) as well as positive and negative predictive values (PPV and NPV) are reported as mean values from 10 simulation runs, after summarizing classification results in a binary form (i.e., order *j* versus all other orders).

Order	Order ID	TPR (%)	TNR (%)	PPV (%)	NPV (%)
*Bunyavirales*	1	24.8	93.7	33.1	90.9
*Caudovirales*	2	41.0	93.3	48.7	92.8
*Herpesvirales*	3	28.9	96.4	51.0	91.6
*Ligamenvirales*	4	56.3	93.6	53.6	94.5
*Mononegavirales*	5	45.1	86.3	31.2	92.8
*Nidovirales*	6	56.6	90.4	46.1	94.4
*Ortervirales*	7	38.3	92.8	41.6	92.3
*Picornavirales*	8	16.4	96.3	36.0	90.2
*Tymovirales*	9	50.3	89.3	40.1	93.6

**Table 6 genes-12-01755-t006:** Mean (minimum, maximum) accuracies achieved by different classifier models to assign a read to the correct viral order in 10 simulation runs.

Model	Mean Accuracy	Min. acc.	Max. acc.
*ANN, 1 hidden layer*	0.40	0.35	0.47
*ANN, 5 hidden layers*	0.41	0.36	0.46
*LDA*	0.22	0.12	0.34
*SVM, linear kernel*	0.27	0.23	0.36
*SVM, radial kernel*	0.36	0.29	0.43
*SVM, polynomial kernel*	0.40	0.33	0.45
*SVM, sigmoid kernel*	0.27	0.24	0.31

## Data Availability

All data used in this study are freely available. Source code used for simulations is available from https://github.com/klausjung-hannover/taxaEstimator, accessed on 3 February 2021.

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
