# Peer review of "Correcting the Estimation of Viral Taxa Distributions in Next-Generation Sequencing Data after Applying Artificial Neural Networks"

_genes, 2021, doi:10.3390/genes12111755_

Round 1

Reviewer 1 Report

Review on the paper 'Correcting the estimation of viral taxa distributions in next-generation sequencing data after applying artificial neural networks'.

The paper proposed a new method that used ANN to classify a virus to different groups at the 'order' level based on sequence data and provided an approach to correct the bias for the reference datasets based on posterior viral taxa distributions. The reviewer agreed that the importance to correct the estimation bias. However, more investigation and method comparisons are needed.

Major comments:
1. The authors used a training set to estimate prior taxa distribution and used an auxiliary set to calculate the conditional classification rate. Why not combing them together, dividing the datasets for training, validation and testing, and then using the validation set for estimating the prior distribution? Please clarify.

2. The authors used the law of the total probability to define the posterior estimation. The posterior estimation should be in a form of summation or integral. But from formula (1), the reviewer did not see that.

3. In the method performance comparison, the authors only compared posterior estimation, prior estimation, and mapping approach. The comparison to other methods from other deep learning and machine learning approach is needed, such as the methods mentioned in the introduction section. Also the authors only provided one deep learning structure in their ANN. How about adding multiple layers and selecting different numbers of nodes in ANN? It's good to see the training process in ANN.

4. The authors need to do more efforts to investigate why the training data distribution matters for the classification in the test set. The authors proposed an approach based on posterior distribution. It will be good to see the authors can explore the reason. It might be good to compare the distributions between training sets and test sets to see how the test sets are different from the training set. Whether the proportions of 9 orders need to be the same in the training set, validation set and test set? Another related question is whether using the current training set is good enough for classifying the test set. It might be possible the test set can learn nothing specifically from the train sets.

5. The authors need to provide quantitative analysis on investigation of the misclassification, if possible. The authors gave some hypotheses on the misclassification in the discussion section. It will be good to see some quantitative comparison. 

6. Another point to further explore is whether there are some ambiguous groups for some virus. In that case, it's hard to assign a clear group to the virus. Is it possible to set a 'unknown' group in ANN?

Minor comment:
It will be good to add a neural network flow plot to Figure 2.  

Author Response

Point-by-point-reply to the reviews on our manuscript

“Kohls et al.: Correcting the estimation of viral taxa distributions in next-generation sequencing data after applying artificial neural networks”

The authors thank both reviewers for the helpful and constructive comments! We have revised our manuscript thoroughly with respect to all points. Please find our answers below.

Reviewer 1:

Review on the paper 'Correcting the estimation of viral taxa distributions in next-generation sequencing data after applying artificial neural networks'.

The paper proposed a new method that used ANN to classify a virus to different groups at the 'order' level based on sequence data and provided an approach to correct the bias for the reference datasets based on posterior viral taxa distributions. The reviewer agreed that the importance to correct the estimation bias. However, more investigation and method comparisons are needed.

Major comments:

  1. The authors used a training set to estimate prior taxa distribution and used an auxiliary set to calculate the conditional classification rate. Why not combing them together, dividing the datasets for training, validation and testing, and then using the validation set for estimating the prior distribution? Please clarify.

Answer: Thanks for this comment! We also considered this idea at the beginning of the project. However, the situation for training the models is special here: training data is not generated from biological samples with a given (limited) sample size. Instead artificial reads are sampled as training data from the viral genomes provided in the NCBI database. This opens the possibility to generate repeatedly additional data (here called the ‘auxiliary data sets’) for testing the trained model and to estimate the conditional class probabilities used for correcting the taxonomic distributions. The validation data in the primary training set and in the auxiliary data sets are only used to train the weights of the neural nets. When merging all the independent auxiliary datasets the variability between these datasets would not be taken into account and we think that the overall accuracy would be judged as too optimistic. We added this argumentation to the discussion section.

  1. The authors used the law of the total probability to define the posterior estimation. The posterior estimation should be in a form of summation or integral. But from formula (1), the reviewer did not see that.

Answer: We added equation (2) which shows, for each test run s,  that the matrix multiplication in equation (1) contains the sum of conditional probabilities multiplied by prior probabilities.

  1. In the method performance comparison, the authors only compared posterior estimation, prior estimation, and mapping approach. The comparison to other methods from other deep learning and machine learning approach is needed, such as the methods mentioned in the introduction section. Also the authors only provided one deep learning structure in their ANN. How about adding multiple layers and selecting different numbers of nodes in ANN? It's good to see the training process in ANN.

Answer: We agree that adding further machine learning approaches is reasonable, and we now did this. In particular, we added results from a more complex ANN with 5 layers and also from SVMs (with different kernels) and linear discriminant analysis. Since none of these models resulted in a better accuracy to classify a read correctly, we left the analysis on taxa frequency correction with only the ANN model with one hidden layer. It was interesting to see, that also an SVM with polynomial kernel performed equally well as the ANN models.

The other deep learning approaches cited in the introduction were designed for different questions and/or their specific structure and input data was not detailed in the original publications. Therefore, a direct comparison with these models was not feasible, here.

  1. The authors need to do more efforts to investigate why the training data distribution matters for the classification in the test set. The authors proposed an approach based on posterior distribution. It will be good to see the authors can explore the reason. It might be good to compare the distributions between training sets and test sets to see how the test sets are different from the training set. Whether the proportions of 9 orders need to be the same in the training set, validation set and test set? Another related question is whether using the current training set is good enough for classifying the test set. It might be possible the test set can learn nothing specifically from the train sets.

Answer: We added this important comparison of different distributions in the data sets to the result section. Specifically, we kept with balanced distributions for training and validation set, i.e. 1/9=11% of training and validation reads from each order. Since the analyst takes these two data sets from a sequence database, there are now restrictions to use unbalanced data sets, here. Of course – and that is the reason for doing meta-genomics – there will be unbalanced scenarios in real world samples. Therefore, we simulated two exemplary scenarios, where the test sets had unbalanced distribution of reads between the nine viral orders. We draw these distributions randomly from the Dirichlet-multinomial distribution in both cases. As can be seen from the new Figure 6, the bias correction approach can still correct for the bias in the prior taxa distribution estimation.

  1. The authors need to provide quantitative analysis on investigation of the misclassification, if possible. The authors gave some hypotheses on the misclassification in the discussion section. It will be good to see some quantitative comparison.

Answer: We added a new Table 6 which shows accuracies for other ML models. As can be seen, a more complex ANN but also an SVM with polynomial kernel result in the same accuracy (but also not better) than the ANN with only one hidden layer. We think that with even more complex models there is rather the risk of overfitting. As you mentioned, we hypothesized in the discussion that also read lengths and input features can be relevant for reducing the miss-classification rate.

  1. Another point to further explore is whether there are some ambiguous groups for some virus. In that case, it's hard to assign a clear group to the virus. Is it possible to set a 'unknown' group in ANN?

Answer: We agree that this would be a reasonable improvement for biological interpretation. However, we think that more detailed research would be necessary to find a good way to implement this. One possibility would be to generate random reads for the training data that don’t belong to any known virus of the nine orders. It is not straightforward how such reads should be composed. Nevertheless, we added the idea to the discussion of our manuscript.

Minor comment:

It will be good to add a neural network flow plot to Figure 2. 

Answer: We agree and added the network flow to this plot.

Reviewer 2:

The manuscript entitles: “Correcting the estimation of viral taxa distribution in next-generation sequencing data after applying artificial neural networks” is the great interest nowadays, because there is not too much information about a standard protocol for understating the taxa distributions from raw data of NGS.

Introduction

Line 26- 29: “These mapping approaches have been proven to be successful in a large number of examples, however, they mostly fail to classify reads from new emerging viruses whose sequences are not yet deposited in a database. While there are diverse tools available for taxonomic classification of NGS reads from novel species, these share several limitations.” I agree with you but you should point out some studies that reported this hypothesis.  You can add some studies based on Kraken-2  or listed on Peabody, for giving more robustness.

Answer: We added the references of Ren et al., Maclot et al., and Parks et al. to the introduction. These papers point at the difficulty to assign reads that belong to novel viruses whose sequences or related genes are not stored in a database, or if even no homologs are available in a database. Further limitations of tools for taxonomic classification were already specified in the introduction text.

Line 71- 79: You should to remove the last paragraph of the introduction, because it is an abstract of subsequently paper.

Answer: It is very common to provide a short outline of the subsequent chapters at the end of the introduction to make it easier for the readership to catch the structure of the paper. However, we agree that our outline was a little but too abstract-like and reduced this section at some points.

In general, the introduction is well explained and written. The issues given by the lack of knowledge are well documented. Also, their approach is well focus, which it is the great interest in the area of metagenomics of newly emerging virus populations.

Materials and Methods

The Table 1 is expendable; you could reference it on the text in order to make easier the lecture. I think that readers who are interested on this information well-known the viruses’ taxa levels.

Answer: We agree and removed this Table 1. Instead, we added a short sentence to the text, together with ref Walker et al. (2019).

Please, modify the table 3, or remove and described on text.

Answer: We also removed Table 3 and put the information into the text.

In summary, the material and methods are well explained in details. Actually, you can follow easily the protocol described. In this sense, this manuscript is of high value for viral metagenomic studies, because the most of research are not at all described, so in this sense we can reproduce the protocol. Nowadays, scientific have problems with sequencing due to the lack of database, but this studies try to improve and share the knowledge of the area until nowadays.

Results

The results are very interesting and the values are well described.  The figure 5 maybe could be improved.

Answer: We increased the size of the symbols and lines in Figure 5 to make it better readable.

Discussion.

In this section, I miss the comparison with others studies, which are represented or seems to be represented the wrong affiliation.

Answer: We included the comparison with other ML models in the results section and mentioned this in the discussion. The different purpose of other studies to use ML for sequence read classification in detailed in the introduction. We again mentioned these differences to our approach in the discussion now.

I miss the conclusion of your study.

Answer: We now added a conclusion section.

In summary, I think that this article is novel and it is of great interest for people who works on the study of virome in all habitats.. and it could help to describe a good protocol to understand which are the gaps in this area in order to don’t described wrong the virome of our samples.

Reviewer 2 Report

The manuscript entitles: “Correcting the estimation of viral taxa distribution in next-generation sequencing data after applying artificial neural networks” is the great interest nowadays, because there is not too much information about a standard protocol for understating the taxa distributions from raw data of NGS.

Introduction

 Line 26- 29: “These mapping approaches have been proven to be successful in a large number of examples, however, they mostly fail to classify reads from new emerging viruses whose sequences are not yet deposited in a database. While there are diverse tools available for taxonomic classification of NGS reads from novel species, these share several limitations.” I agree with you but you should point out some studies that reported this hypothesis.  You can add some studies based on Kraken-2  or listed on Peabody, for giving more robustness.

Line 71- 79: You should to remove the last paragraph of the introduction, because it is an abstract of subsequently paper.

In general, the introduction is well explained and written. The issues given by the lack of knowledge are well documented. Also, their approach is well focus, which it is the great interest in the area of metagenomics of newly emerging virus populations.

 Materials and Methods

The Table 1 is expendable; you could reference it on the text in order to make easier the lecture. I think that readers who are interested on this information well-known the viruses’ taxa levels.

Please, modify the table 3, or remove and described on text.

In summary, the material and methods are well explained in details. Actually, you can follow easily the protocol described. In this sense, this manuscript is of high value for viral metagenomic studies, because the most of research are not at all described, so in this sense we can reproduce the protocol. Nowadays, scientific have problems with sequencing due to the lack of database, but this studies try to improve and share the knowledge of the are until nowadays.

Results

The results are very interesting and the values are well described.  The figure 5 maybe could be improve.

Discussion.

In this section, I miss the comparison with others studies, which are represented or seems to be represented the wrong affiliation.

I miss the conclusion of your study.

In summary, I think that this article is novel and it is of great interest for people who works on the study of virome in all habitats.. and it could help to describe a good protocol to understand which are the gaps in this area in order to don’t described wrong the virome of our samples.

Author Response

(The authors gave the same response as above.)
